# Advanced genomics identifies growth effectors for proteotoxic ER stress recovery in *Arabidopsis thaliana*

Dae Kwan Ko [iD] [1,2] & Federica Brandizzi [iD] [1,2,3 ✉]

Adverse environmental and pathophysiological situations can overwhelm the biosynthetic capacity of the endoplasmic reticulum (ER), igniting a potentially lethal condition known as ER stress. ER stress hampers growth and triggers a conserved cytoprotective signaling cascade, the unfolded protein response (UPR) for ER homeostasis. As ER stress subsides, growth is resumed. Despite the pivotal role of the UPR in growth restoration, the underlying mechanisms for growth resumption are yet unknown. To discover these, we undertook a genomics approach in the model plant species *Arabidopsis thaliana* and mined the gene reprogramming roles of the UPR modulators, basic leucine zipper28 (bZIP28) and bZIP60, in ER stress resolution. Through a network modeling and experimental validation, we identified key genes downstream of the UPR bZIP-transcription factors (bZIP-TFs), and demonstrated their functional roles. Our analyses have set up a critical pipeline for functional gene discovery in ER stress resolution with broad applicability across multicellular eukaryotes.

[1] MSU-DOE Plant Research Lab, Michigan State University, East Lansing, MI, USA. [2] Great Lakes Bioenergy Research Center, Michigan State University, East Lansing, MI, USA. [3] Department of Plant Biology, Michigan State University, East Lansing, MI, USA. ✉email: fb@msu.edu

Specialized ER membrane-associated sensors perceive ER stress and control growth in physiological conditions and ER stress-causing conditions. The ribonuclease and kinase inositol requiring enzyme 1 (IRE1) is the most conserved ER stress sensor[1,2]. IRE1 splices the mRNA of a bZIP-transcription factor (TF) (*Xbp1* in mammals[3–5], *bZIP60* in plants[6], and *Hac1* in yeast[7]). The spliced bZIP-TF mRNA is then translated into an active TF that modulates the expression of UPR genes in the nucleus[8–10]. In yeast, the UPR depends solely on Ire1p; however, in metazoans and plants, the UPR has expanded to additional arms and components within each arm[11–14], likely to suit the complexity of tissue development and growth of multicellular eukaryotes. The additional UPR arm conserved in metazoans and plants relies on ER membrane-tethered bZIP-TFs, namely the Activating Transcription Factor 6 (ATF6) in metazoans[15] and bZIP28 in plants[1]. Upon stress, these bZIP-TFs are transported to the Golgi apparatus for proteolytic activation and release of the TF domain, which is then translocated into the nucleus to control UPR gene expression[16,17]. The UPR bZIP-TFs bind to promoters of UPR target genes via ER stress-responsive *cis*-regulatory elements (CREs), which are generally conserved[15,18–20]. The DNA binding of the UPR bZIP-TFs can also be modulated by other TFs and transcriptional regulators[21–23], underscoring an elevated complexity in the regulation of UPR gene expression.

In the early phase of ER stress responses, the ER stress sensors activate the adaptive phase of the UPR, which involves the production of ER chaperones and foldases to restore the biosynthetic capacity of the ER[2,24]. During ER stress, growth is reduced[25–27], but, as ER stress is mitigated, growth is resumed[28–30]. Successful restoration of growth during ER stress recovery requires an intact UPR. Indeed, during ER stress recovery, the loss of either bZIP28 or bZIP60 partially hampers organ growth, specifically the root[30–32]. Noticeably, either bZIP28 or bZIP60 is not essential for seedling growth in physiological conditions nor in persistent ER stress conditions[23,30–32]. Therefore, these UPR bZIP-TFs are individually necessary for growth during ER stress recovery. Although genetic evidence poses that the growth mechanisms controlled by bZIP28 and bZIP60 in ER stress recovery are partially overlapping[30], their identity is yet completely unknown. The identification of the effectors that control growth in dependence of the UPR bZIP-TFs would contribute to the understanding of the interplay of stress responses and the growth of multicellular organisms at a mechanistic level. In this work, we addressed this knowledge gap and focused on identifying and characterizing the downstream effectors of the UPR bZIP-TFs in organ growth during ER stress recovery through innovative integration of large-scale time-series RNA sequencing (RNA-seq), chromatin immunoprecipitation sequencing (ChIP-seq), and cistrome analyses into gene regulatory network modeling in the model plant species *Arabidopsis thaliana* (hereafter Arabidopsis).

## Results

**bZIP28 and bZIP60 are required for root growth during recovery from ER stress**. As a first step to determine growth effectors in ER stress resolution, we sought to set up experimental conditions of recovery from ER stress caused by tunicamycin (Tm), a widely used ER stress-inducing agent, which is highly effective in plant cells[33]. We used DMSO (Tm-solvent) as mock control. We exposed Arabidopsis Col-0 seedlings to Tm for 6 h and then either transferred them to growth media containing no Tm for ER stress recovery or continuously exposed them to Tm to elicit adaptive ER stress responses (Fig. 1a). Using a high-resolution LC/MS/MS on our samples, we verified a significant decrease of Tm levels during the ER stress recovery time-course (75.4% decrease at 24 h of recovery compared to 0 h) and

increased levels of Tm in adaptive ER stress conditions (Fig. 1b). In the progression of ER stress recovery, the levels of induction of spliced *bZIP60* (*sbZIP60*) transcripts, a well-established biomarker for UPR activation[10,25], were substantially attenuated (Fig. 1c). Conversely in adaptive ER stress conditions, the *sbZIP60* levels were only mildly reduced at 12 and 24 h compared to the 6 h of Tm treatment. These results indicate that we successfully set up the experimental conditions for a time-course of ER stress recovery in which the UPR is significantly attenuated. Based on these experimental premises, we quantitatively followed the growth of Col-0 and single and double loss-of-function mutants of the UPR bZIP-TFs (i.e., *bzip28-2*[16], *bzip60-2*[10], and *bzip28-2 bzip60-1*[34]) in an extended time-course of ER stress recovery (Fig. 1d, e and Supplementary Fig. 1a, b). While *bzip28-2 bzip60-1* did not show growth recovery, consistent with previous findings[21,30], *bzip28-2* and *bzip60-2* reestablished growth: the total biomass of *bzip60-2* was 72% compared to Col-0, and was significantly higher than that of *bzip28-2* and *bzip28-2 bzip60-1*. We observed a significantly higher relative growth rate (Tm/DMSO) of the primary root of *bzip60-2* compared to *bzip28-2*, yet lower compared to Col-0. In physiological conditions, these mutants did not show significant differences in root growth, while *IRE1* loss-of-function mutations showed significant root growth defects (Supplementary Fig. 1c, d), as reported earlier[26,34]. These results support the notion that bZIP28 and bZIP60 are primarily required for the actuation of partially overlapping genetic pathways for root growth in ER stress recovery conditions[30–32].

**bZIP28 and bZIP60 initiate broad gene reprogramming in ER stress recovery**. We next aimed to identify the downstream effectors of bZIP28 and bZIP60 in growth restoration during ER stress resolution using a genomics approach. We reasoned that the initial phase of ER stress recovery would be critical to actuate transcriptional regulations by bZIP28 and bZIP60 to restore organ growth. Therefore, we performed RNA-seq analyses in Col-0, *bzip28-2*, and *bzip60-2* collected at 0, 12, and 24 h of ER stress recovery, using the corresponding DMSO-treated samples as additional controls in each genotype at each time-point (Supplementary Data 1). From a highly reproducible dataset (Supplementary Fig. 2a, b), we identified differentially expressed genes (DEGs) in each genotype at each time-point (Fig. 2a and Supplementary Data 2). The number of DEGs was relatively low at 0 h (576, Col-0; 629, *bzip28-2*; 552, *bzip60-2*) but substantially increased at 12 h, and was significantly higher in both *bzip28-2* (4076; $P = 1.3 \times 10^{-10}$) and *bzip60-2* (4254; $P = 3.2 \times 10^{-10}$) compared to Col-0 (2484). Although the number of DEGs (2930) in Col-0 increased at 24 h compared to 12 h, it decreased in *bzip28-2* (3162) and in *bzip60-2* (2316) at 24 h to significantly different levels compared to Col-0 (*bzip28-2*, $P = 6.7 \times 10^{-15}$; *bzip60-2*, $P = 6.6 \times 10^{-15}$). About 20% of the DEGs were common to the *bzip28-2* and *bzip60-2* backgrounds and 6–7% were unique to each background (Supplementary Fig. 2c).

We observed that some of the UPR biomarker genes were regulated in a genotype-dependent manner (Supplementary Fig. 2d–f). For example, while substantially induced in all genotypes, the expression induction of *ER-resident J protein 3B* (*ERdj3B*) was significantly attenuated in *bzip28-2*. Therefore, we next performed genome-scale pairwise comparisons (two-tailed Student's *t*-test, $P < 0.01$) of expression Log$_2$FC (Tm/DMSO) of each DEG (a total of 6670 genes) at each time-point across the genotypes (Col-0 vs. *bzip28-2*, Col-0 vs. *bzip60-2*, and *bzip28-2* vs. *bzip60-2*) (Supplementary Data 2). Overall, we identified 875 and 904 DEGs regulated in a bZIP28- and the bZIP60-dependent manner in at least one time-point, respectively (Fig. 2b). In addition, 362 DEGs were significantly different between *bzip28-2*

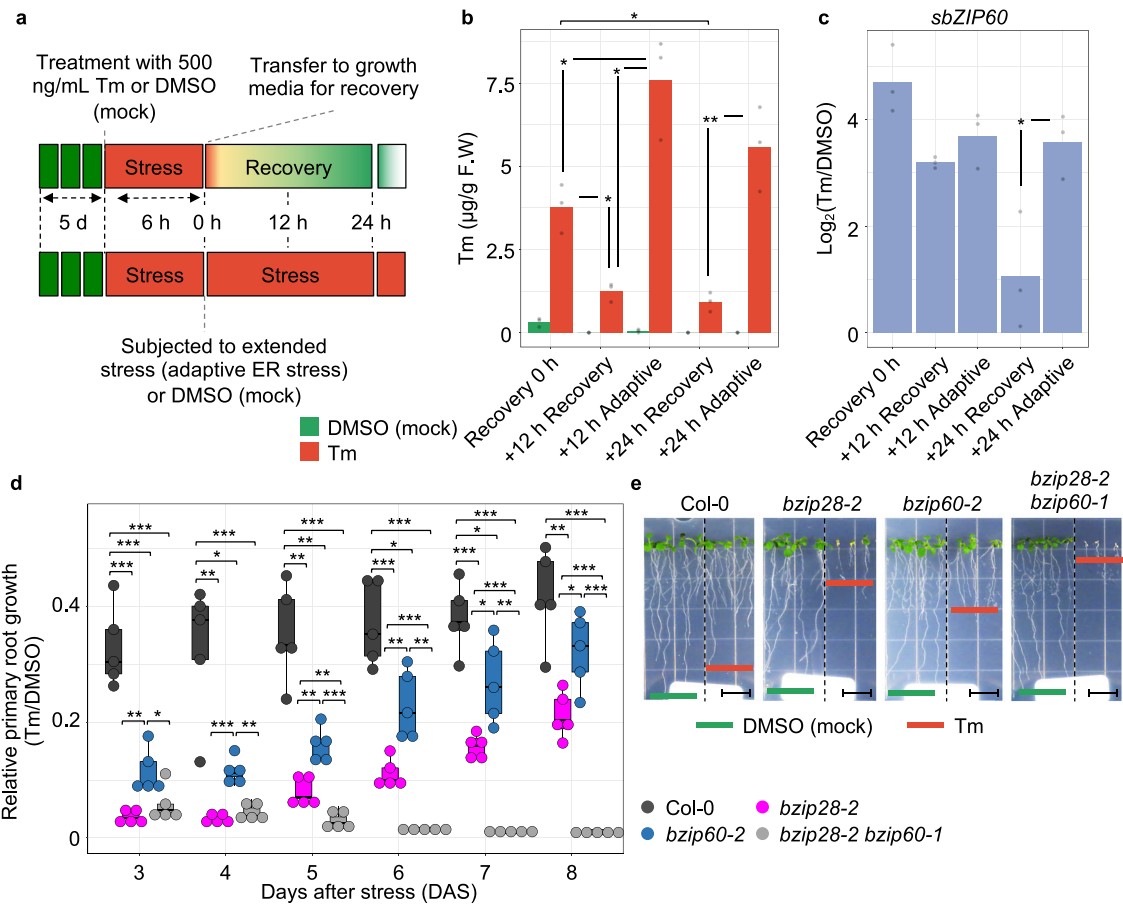

**Fig. 1 Characterization of Arabidopsis responses to ER stress recovery. a** Scheme of the experimental design for ER stress recovery assay. Seedlings were germinated in normal growth conditions for 5 days and then treated for 6 h with either Tm or DMSO (mock control). Following the drug treatment, seedlings were transferred to drug-free growth media to allow ER stress to subside and growth to restore. **b** Quantitative LC/MS/MS analysis of Tm contents in Col-0 whole seedlings subjected to 6 h of Tm (or DMSO) treatment followed by ER stress recovery or extended ER stress (adaptive ER stress) as described in (**a**). $n$ (number of biological replicates) = 3 (~50 seedlings per replicate). **c** Relative expression levels of *sbZIP60* (Log$_2$[Tm/DMSO]) at each time-point under each condition. $n$ = 3 (12 seedlings per replicate). **d** Relative growth rate of the primary root of Col-0, *bzip28-2*, *bzip60-2*, and *bzip28-2 bzip60-1* in ER stress recovery as described in (**a**). The relative growth rate of the primary root was monitored during ER stress recovery. $n$ = 5 (6 seedlings per replicate). **e** Representative images of the primary root of Col-0, *bzip28-2*, *bzip60-2*, and *bzip28-2 bzip60-1* in ER stress recovery taken at 8 days in recovery from ER stress. Scale bar (black line) = 1.4 cm. *$P$ < 0.05, **$P$ < 0.01, ***$P$ < 0.001 (two-tailed Student's $t$-test). The source data of **b**–**d** is provided in Supplementary Data 6.

and *bzip60-2*. Hereafter, we called these genotype-dependently regulated DEGs as differentially-differentially expressed genes (DDEGs). Intriguingly, the number of DDEGs was relatively small at 0 h (50, bZIP28-dependent; 52, bZIP60-dependent), but, for bZIP28 and bZIP60 respectively, it increased to 645 and 653 at 12 h and declined to 216 and 218 at 24 h. Thus, the responses of genes to the lack of functional bZIP28 or bZIP60 followed a temporal and partially non-overlapping profile during ER stress recovery.

The observed gene responses were most likely controlled by activities of two canonical CREs, the ER stress-responsive element-I (ERSE-I)[15] and unfolded protein response element-I (UPRE-I)[18], as supported by the evidence that these CREs were predominantly enriched in the promoters of DDEGs compared to the other UPR CREs (i.e., ERSE-II[35], UPRE-II[20], and UPRE-III[12]) (Fig. 2c). Through *de novo* motif analyses, we also identified a variety of TF motifs temporally enriched in the gene promoters (Fig. 2d). For example, the binding motif (ACAAAAAAAA) of C2H2 Zinc Finger TF Indeterminate (ID)-Domain11 (IDD11) was highly enriched in promoters of both bZIP28- and bZIP60-dependently regulated genes (i.e., Col-0 vs. *bzip28-2*; Col-0 vs. *bzip60-2*) at 12 h, indicating that IDD11 could directly bind to

promoters of these DDEGs. Also, the binding motif (5′-CCGGCG-3′) of an ERF/AP2 TF (AT3G58630) was exclusively enriched in promoters of bZIP28-dependently regulated genes while the binding motif (5′-GAAGAAGAAGA-3′) of NTM1-LIKE 8 (NTL8), a NAM, ATAF1/2, and CUC (NAC) TF, was enriched in the promoter of bZIP60-dependently regulated genes at 12 h. The enrichment of these CREs in the gene promoters at 12 h strongly suggests that the regulatory dynamics of bZIP28 and bZIP60 may be associated with activities of other TFs that could operate as coregulators or downstream of bZIP28 and/or bZIP60 by bindings to non-canonical UPR CREs.

To funnel the results obtained thus far towards the identification of genes involved in organ growth in dependence of bZIP28 and/or bZIP60 during ER stress recovery, we performed a weighted gene coexpression network analysis (WGCNA)[36], which categorized the 6670 DEGs based on expression similarity across time-points and genotypes (Fig. 3a). This approach enabled us to identify 14 coexpression modules that were color-coded and ranged in size from 25 (light green) to 890 genes (turquoise). Then, to identify the biological functions of each coexpression module we performed Gene Ontology (GO) enrichment analysis and compared the GO categories enriched in each coexpression

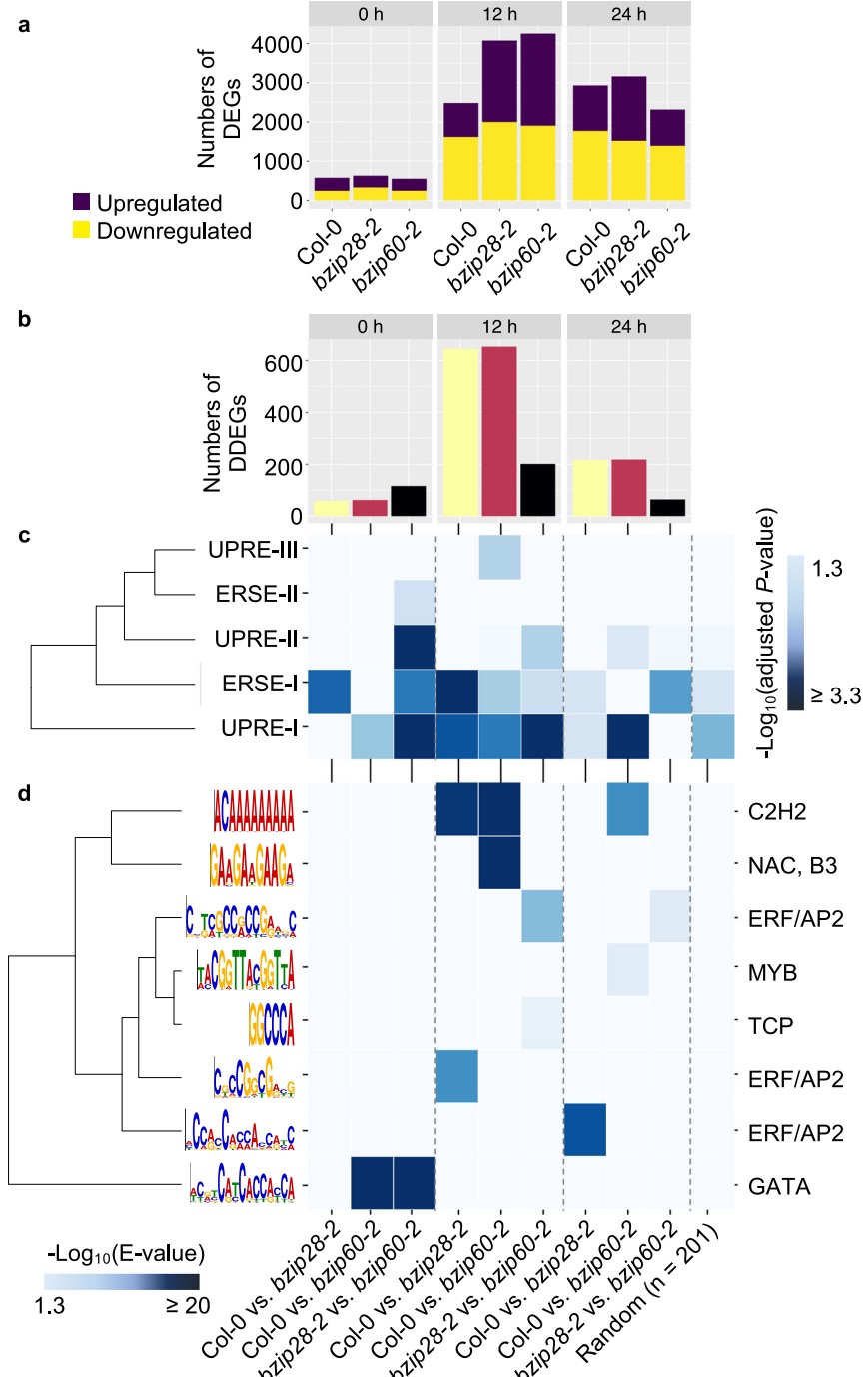

**Fig. 2 Temporal differential gene regulation in a genotype-dependent manner and cistrome profiling in ER stress recovery. a** Numbers of DEGs in each genotype at each time-point (|Log2FC| > 1, Adjusted $P$ > 0.05). Statistical differences of the number change between each mutant and Col-0 were calculated by Fisher's exact test and indicated in the main text. **b** Numbers of DDEGs at each time-point. **c, d** Temporal dynamic enrichments of CREs on 1-kb promoters of DDEGs. The upper heatmap (**c**) shows enrichments of the canonical ER stress-responsive CREs on the promoters of DDEGs. ERSE-I, CCAAT-N$_{10}$-CACG. ERSE-II, ATTGG-N$_2$-CACG. UPRE-I, TGACGT-GG/A, UPRE-II, GATGACGCGTAC. UPRE-III, TCATCG. The lower heatmap (**d**) shows enrichments of de novo motifs on the promoters of DDEGs. The motifs are displayed on the left of each row. The best match to a transcription factor family for each motif is displayed to the right of each row. The median number of DDEGs across all pairwise comparisons is 201, and we randomly selected 201 DEGs and included the analysis as a control set. This analysis revealed multiple TF motifs significantly enriched. The source data is provided in Supplementary Data 6.

module (Fig. 3b and Supplementary Data 3). Each coexpression module exhibited strong enrichment of distinct GO categories associated with important biological processes such as metabolic pathways, hormone responses, and stress responses, pointing to an extensive level of the biological relevance of gene regulation in

the UPR. Among them, the pink module (354 genes) showed a unique expression profile of the eigengene in each genotype (i.e., increased expression in Col-0 over time but variable expression in *bzip28-2* and *bzip60-2* at different levels), and importantly, it exhibited exclusive enrichment of root development-related

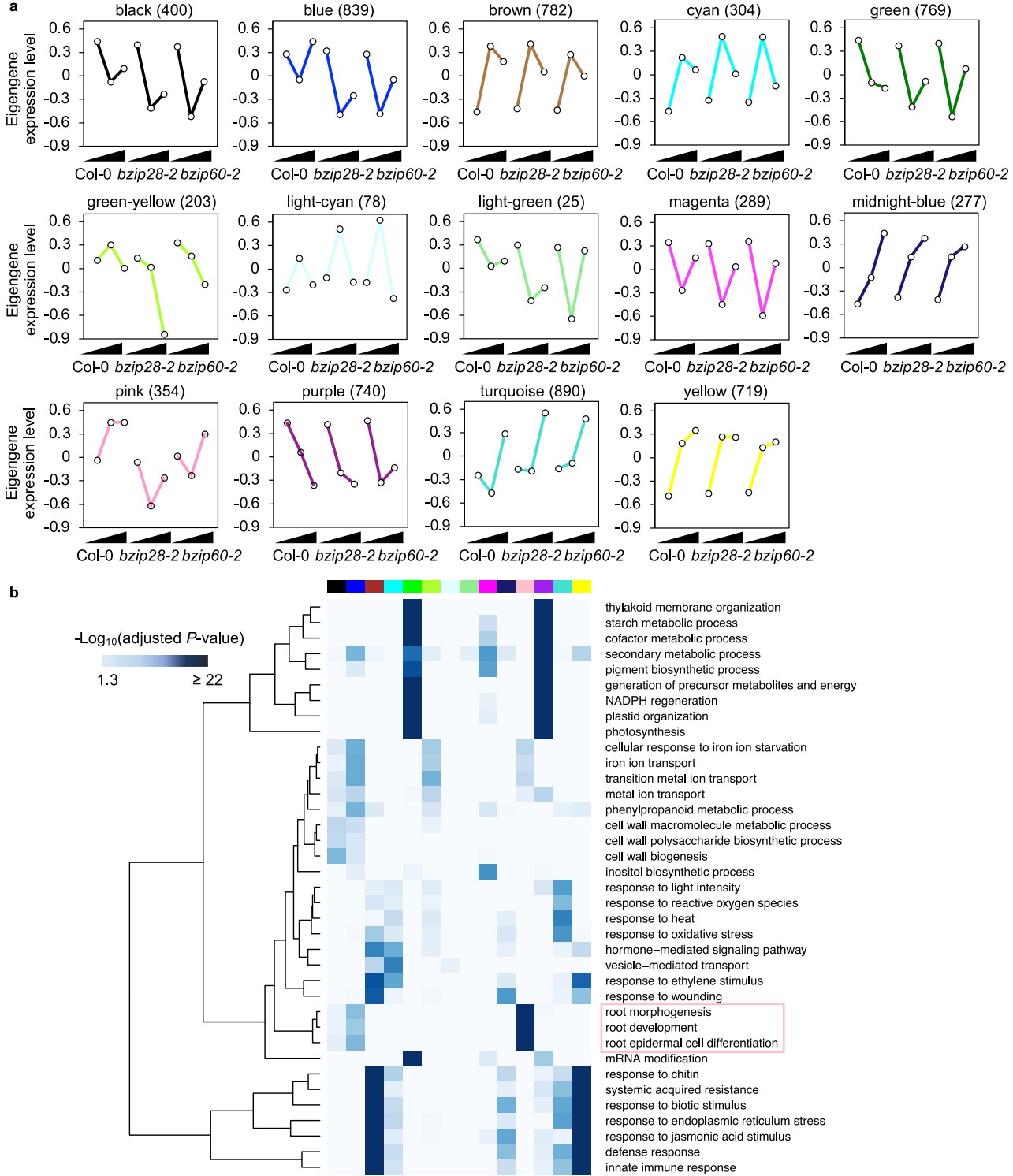

**Fig. 3 Coexpression analysis generates modules of DEGs coexpressed across genotypes and time-points of ER stress recovery. a** All DEGs identified were subjected to WGCNA to detect modules of coexpressed genes based on the expression similarity. Each graph shows the expression of the module eigengene, which can be considered as the representative expression in the respective coexpression module. The y-axis indicates $Log_2$ expression values relative to the mean expression across all samples. The x-axis indicates time-points in each genotype. The modules are named according to their color created through WGCNA. The number of genes in each module is in parentheses. **b** GO analysis of genes in each coexpression module (adjusted $P < 0.05$, Hypergeometric test). The respective module is indicated by the color bar above the heatmap. GO terms strongly enriched in genes of the pink module are marked by a pink box. A full list of GO terms is provided in Supplementary Data 3. The source data is provided in Supplementary Data 6.

pathways (Fig. 3b). The genotype-specific expression of the eigengene and exclusive enrichment of the GO terms associated with root development (several of the annotations have been experimentally validated[12,37,38]) in the pink model are in the line with the non-completely overlapping growth phenotype of the primary root in *bzip28-2* and *bzip60-2* (Fig. 1d, e and Supplementary Fig. 1b). Therefore, we decided to focus on the pink module for functional characterization.

**Gene regulatory network construction based on coexpression networks coupled with DNA-binding profiles.** We next aimed to establish the in vivo temporal DNA-binding interactomes of bZIP28 and bZIP60 as a strategy to identify direct targets of these TFs in ER stress recovery. To do so, we performed ChIP-seq analyses at 0, 12, and 24 h of ER stress recovery in stable transgenic lines (Col-0 background) expressing a green fluorescent protein (GFP)-tagged fusion to either bZIP28 (*35Spro:GFP-bZIP28*/Col-0)[39] in which the transcripts of *bZIP28* were not overexpressed compared to Col-0 (rather constitutively expressed in both mock and Tm treatments) or a bZIP60 in *bzip60-2* (*bZIP60pro:GFP-bZIP60*/*bzip60-2*)[40], which successfully rescued the root the phenotype and gene expression profiles of *bzip60-2* (Supplementary Fig. 3a–d). ChIP-quantitative PCR (ChIP-qPCR) results showed that GFP-bZIP28 and GFP-bZIP60 bound to ERSE-I[15,18], which is present in the promoter regions of *BiP3* and *ERdj3B* (fragments A and B; Supplementary Fig. 3e, f) specifically in Tm treatment, but not in DMSO treatment. This observation is in line with the notion that bZIP28 and bZIP60 operate in the first layer of the UPR upon ER stress-specific protease cleavage[16] and unconventional splicing by IRE1[10], respectively, for activation as TFs. However, these TFs did not bind to an *Erdj3B* promoter region without the ERSE-I (fragment C) or in the ChIP mock controls (i.e., no antibody). These DNA-binding activities were unlikely to be driven by the GFP tag. This is supported by the evidence from ChIP assays in a transgenic line expressing a yellow fluorescent protein (YFP, a variant of GFP with three-point mutations in the chromophore)[41]. Specifically, YFP was enriched on the three loci only at background levels despite a constitutive *YFP* expression, which was 42 times higher than the expression of endogenous *sbZIP60* in Tm-treated conditions (Supplementary Fig. 3f, g). Although *bZIP28* was expressed at similar levels in *35Spro:GFP-bZIP28*/Col-0 and Col-0 (Supplementary Fig. 3a), we could not exclude a possibility that the constitutive expression of *bZIP28* may lead to some spurious binding of bZIP28 to the genome. Thus, we subjected our results to a rigorous filtering protocol using an analysis pipeline developed based on existing guidelines[42,43] with a key modification: any peaks found in both Tm-treated samples and the corresponding DMSO-treated samples, which were processed in parallel, were removed (see Methods). This step was added in the analysis pipeline to ensure that the resulting bZIP28 binding peaks were specific to Tm treatment. Using this approach, we captured a high-confident set of 676 and 467 UPR-specific peaks of bZIP28 and bZIP60, respectively, the majority of which were located in the proximal gene promoters (i.e., within 1-kb of the transcription start site) (Supplementary Fig. 4a, b), consistent with the binding behavior of other TFs in Arabidopsis[44]. We observed temporal changes of bZIP28- and bZIP60-binding signals (Input-normalized ChIP-seq signals), whereby an initial binding peak at 0 h was followed by a substantial decline at 12 h, and another peak at 24 h of ER stress recovery (Supplementary Fig. 4c, d). *De novo* motif analysis identified that ERSE-I was one of the top-scoring motifs (E-value $= 3.62 \times 10^{-169}$ in bZIP28 UPR-specific binding peaks; $E$ value $= 7.1 \times 10^{-103}$ in bZIP60 UPR-specific binding peaks), and that ERSE-I was centered at the

summit of these binding peaks (Supplementary Fig. 4e, f), validating the high quality of our data. Intriguingly, the analysis also found other TF-binding motifs exclusively enriched in either bZIP28 or bZIP60 UPR-specific binding peaks (Supplementary Fig. 4g–i). The results support the notion that ERSE-I is the canonical ER stress-responsive CRE acting for DNA-binding activities of both bZIP28 and bZIP60[6,19], but also reveal potential coregulators exclusive to either bZIP28 or bZIP60 in ER stress recovery. We then mapped bZIP28 and bZIP60 UPR-specific binding peaks to the potential target genes and found a total of 440 bZIP28- and 356 bZIP60-bound genes (Supplementary Fig. 4g and Supplementary Data 4). Among these genes, 162 genes were bound by both bZIP28 and bZIP60 (significant overlap validated by hypergeometric test, $P = 7.7 \times 10^{-203}$), including *BiP3* and *Erdj3B*, and showed a strong enrichment of GO terms associated with ER stress. Interestingly, the genes exclusively bound by bZIP28 were enriched with GO terms associated with biotic stress-related processes. In contrast, the genes exclusively bound by bZIP60 were enriched with GO terms associated with abiotic stress-related processes (Supplementary Data 5). Overall, these results are consistent with a partially overlapping function of bZIP28 and bZIP60 in ER stress recovery. Importantly also, they provide genome-scale evidence that bZIP28 and bZIP60 not only bind to known genes involved in ER stress responses but also that they interact with genes involved in non-completely overlapping stress pathways during ER stress recovery.

To couple the calculated coexpression network connectivity (i.e., the strength of interaction) with a regulatory relationship to bZIP28 and bZIP60, the DNA-binding profiles of bZIP28 and bZIP60 were integrated into the pink module, revealing that the pink module consists of two subnetworks (hereafter named as pink-sub1 and pink-sub2) (Fig. 4a). In pink-sub1, bZIP28 and bZIP60 bound only to the promoters of three genes and one gene, respectively, supporting the possibility that the sub-module includes 239 genes indirectly regulated by bZIP28 and bZIP60. In contrast, 25% of genes in pink-sub2 were bound exclusively by bZIP60, supporting the likelihood of a higher level of direct regulation by bZIP60 of the genes in this sub-module compared to pink-sub1. Therefore, the ChIP-seq results support the RNA-seq data that, although the entire pink module is controlled by both UPR bZIP-TFs, each of these bZIP-TF contributes to controlling the expression of the genes in each subnetwork to a different extent (Fig. 3a). Consistent with these observations, most genes in pink-sub1 were downregulated while pink-sub2 genes were predominantly upregulated during ER stress recovery (Fig. 4b), supporting distinctive gene regulation signature between pink-sub1 and pink-sub2. Then to experimentally characterize pink-sub1 and pink-sub2, as a first step we used a stringent protocol to predict the regulatory-hub genes (i.e., the genes that are most connected with other genes in a network)[45] in each subnetwork. We considered two main criteria: ranking in the top ≥10% of network degree (i.e., the number of connections) and having a known phenotype of loss-of-function mutations or expected role in roots, cell development, and/or ER stress processes. According to these criteria, out of the genes in the top ≥10% of network degree (out of 242 genes used for the network construction), we selected *Can of Worms1* (*COW1*) (ranked in the top 4.0%), which encodes a phosphatidylinositol transfer protein essential for root hair tip growth[37,46], and root cell wall formation-related *Leucine-rich repeat/Extensin1* (*LRX1*)[37,47] (ranked in the top 1.6%), as major hub genes in pink-sub1. We also selected the *Integral membrane Yip1 family protein 1a* (*YIP1a*), whose yeast and mammalian homologs function in ER-Golgi transport[48,49], with the highest network degree in pink-sub2 (out of 43 genes used for the network

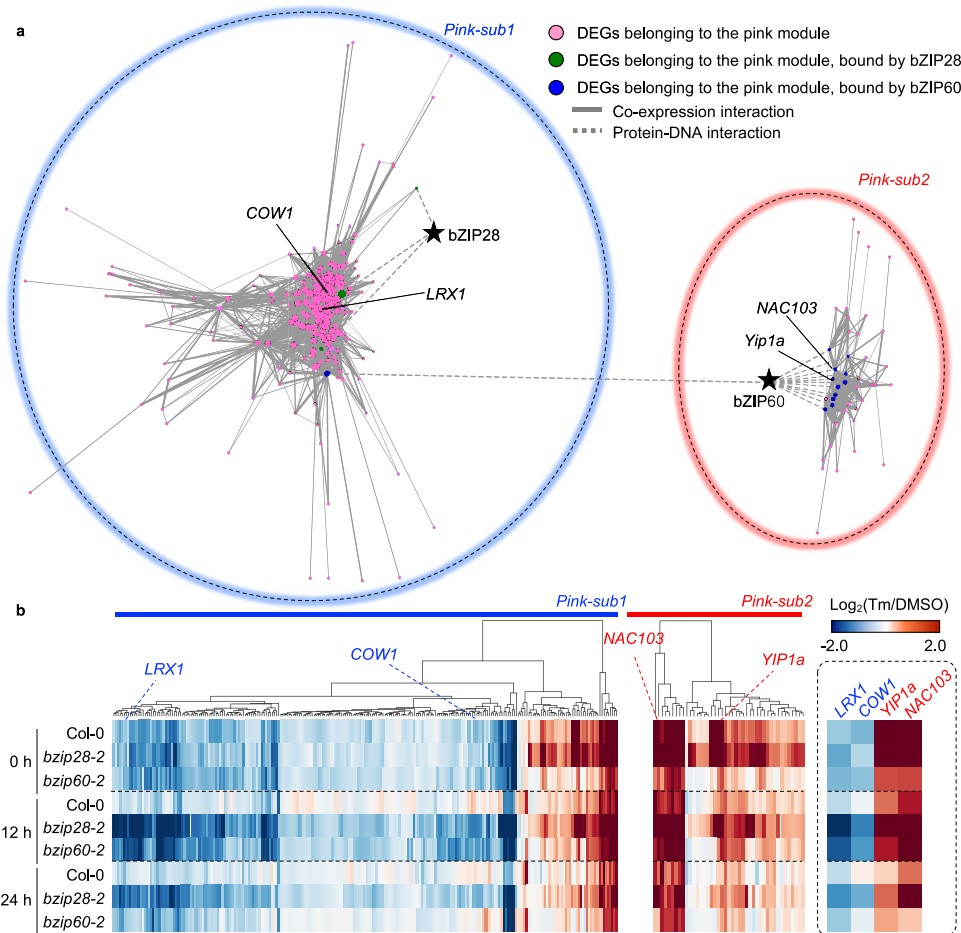

**Fig. 4 A gene regulatory network of bZIP28 and bZIP60 underlying root growth recovery from ER stress. a** A gene regulatory network in the pink module was constructed based on the coexpression network connectivity coupled with the ChIP-seq data of bZIP28 and bZIP60. The node size indicates the degree of coexpression connections. Edge thickness indicates network weight, which indicates connection strength. Pink-sub1 and 2 were indicated by blue and red circles, respectively. **b** Expression Log₂FC of genes that belong to pink-sub1 or pink-sub2 in ER stress recovery. Genes were hierarchically clustered on the expression pattern. Candidates of regulatory-hub genes are indicated in the heatmaps. The source data of **b** is provided in Supplementary Data 6.

construction). Noticeably in pink-sub2, our ChIP-seq analysis identified a known root growth-related direct target of bZIP60, *NAC-domain containing protein103* (*NAC103*)[12] (ranked in the top 9.3%), as being bound by bZIP60 (Supplementary Data 4), lending support to the validity of our approach.

**Functional characterization of the gene regulatory network in ER stress recovery.** Next, except for *NAC103* for which a null T-DNA allele is not available, we isolated the respective loss-of-function T-DNA mutants of *COW1* (*cow1-3*), *LRX1* (*lrx1-4*), and *Yip1a* (*yip1a-1* and *yip1a-2*), which we confirmed being null alleles (Supplementary Fig. 5a–d). We then performed ER stress recovery assays to measure the relative growth (Tm-treated/DMSO-treated) of the primary root (Fig. 5a). In physiological conditions of growth, the primary root growth of these mutants and Col-0 was similar (Supplementary Fig. 6). In conditions of ER stress recovery, while *lrx1-4* showed no growth alteration, *cow1-3*, *yip1-1*, and *yip1a-2* exhibited a significant reduction in the relative growth rate of the primary root compared to Col-0 (Fig. 5a). Although the reduction was not as severe as *bzip28-2* or *bzip60-2*, likely due to functional redundancy with other genes within the pink module, these results experimentally validate that, among the pink module genes, *COW1*, which is bound by neither bZIP28 or bZIP60 and *YIP1a* which is bound exclusively by

bZIP60, have an important role in root growth in recovery from ER stress. We next aimed at consolidating these observations by testing an epistatic relationship of *COW1* and *YIP1a* with *bZIP28* and *bZIP60* in ER stress recovery. Therefore, we generated the following high-order mutants: *cow1-3 bzip28-2*, *cow1-3 bzip60-2*, *yip1a-1 bzip28-2*, *yip1a-1 bzip60-2*, and *cow1-3 bzip28-2 bzip60-1*. The *yip1a-1 bzip28-2 bzip60-1* triple mutant was not recovered due to lethality. We tested the ability of the high-order mutants to reestablish root growth in recovery from ER stress. Consistent with an epistatic relationship with bZIP28 and bZIP60, we found that *cow1-3 bzip28-2* and *yip1a-1 bzip28-2* displayed a significantly higher relative root growth recovery compared to *bzip28-2*. Furthermore, the growth defect of *bzip60-2* was suppressed by the *cow1* mutation (*cow1-3 bzip60-2*) and *yip1a* mutation (*yip1a-1 bzip60-2*). A direct comparison of the genetic effects between *cow1-3* and *yip1a-1* on pink-sub1 and pink-sub2, respectively, is not possible because the relative root growth defect of *bzip60-2* (Tm-treated/DMSO-treated, 0.204) is not as severe as *bzip28-2* (Tm-treated/DMSO-treated, 0.076) and the network size of pink-sub2 (43 genes) is substantially smaller than pink-sub1 (242 genes). Nonetheless, our epistasis results are consistent with our network predictions that bZIP28 is more tightly associated with pink-sub1, in which *COW1* appears as a regulatory-hub, compared to pink-sub2, and that pink-sub2 depends largely on bZIP60 (Fig. 4a).

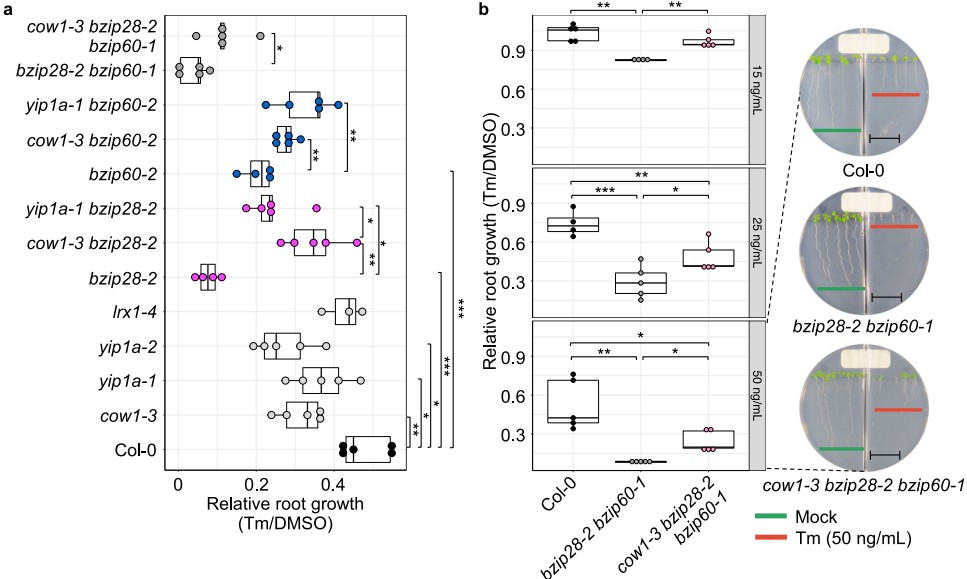

**Fig. 5 Genetic interactions of mutant alleles of regulatory-hub genes with *bzip28-2*, *bzip60-2*, and *bzip28-2 bzip60-1*. a** Relative growth of primary root of T-DNA mutants of the selected regulatory-hub gene candidate (*cow1-3*, *yip1a-1*, *yip1a-2*, and *lrx1-4*) and their higher-order mutants crossed with *bzip28-2*, *bzip60-2*, or *bzip28-2 bzip60-1* in ER stress recovery as described in Fig. 1a. $n = 3–5$ (6 seedlings per replicate). **b** Suppression of *bzip28-2 bzip60-1* by *cow1-3* in chronic ER stress. Seeds germinated directly on the growth media in split plates of which one half contained Tm (15, 25, or 50 ng/mL) while the other half contained DMSO only (the corresponding mock control). The relative growth rate was measured 11 days after growth. $n = 4–5$ (3–5 seedlings per replicate). Scale bar (black line) = 1.2 cm. In boxplots, each box is bounded by the lower and upper quartiles from which the whisker extends 1.5× the interquartile range, and central bars represent the medians. Representative pictures of 50 ng/mL Tm plates are shown at the right. *$P < 0.05$, **$P < 0.01$, ***$P < 0.001$ (two-tailed Student's *t*-test). The source data is provided in Supplementary Data 6.

Because *cow1-3* recovered the root growth phenotype linked to the individual loss of bZIP28 and bZIP60 in ER stress recovery conditions, we next aimed to test whether the loss of *COW1* could recover the sub-lethal phenotype of the *bzip28-2 bzip60-1* double mutant. We found that the *cow1-3 bzip28-2 bzip60-1* triple mutant partially recovered the sub-lethality of *bzip28-2 bzip60-1* mutant in ER stress conditions, further implying that COW1 functions in regulating root growth through the process regulated by bZIP28 and bZIP60 (Fig. 5a). We then extended these findings to test the role of COW1 in chronic ER stress conditions (direct germination and growth on Tm-containing growth media) (Fig. 5b). In chronic ER stress, bZIP28 and bZIP60 redundantly regulate survival strategies (i.e., normal growth in *bzip28-2* and *bzip60-2*, but lethality in *bzip28-2 bzip60-1*)[13,30]. Consistently, the relative growth rate of *bzip28-2 bzip60-1* primary root was significantly reduced compared to Col-0 under all conditions tested, including 50 ng/mL Tm-containing media on which the root growth of *bzip28-2 bzip60-1* was stunted soon after germination. Conversely, *cow1-3 bzip28-2 bzip60-1* showed significantly higher relative root growth than *bzip28-2 bzip60-1* in these conditions, demonstrating a remarkable genetic suppression of the *bzip28-2 bzip60-1* ER stress phenotype by *cow1-3* also in chronic ER stress. Taken together, our network analysis functionally characterized the temporal dynamics of gene regulation by bZIP28 and bZIP60 for reestablishing root growth in ER stress recovery and identified key effector genes acting, directly and indirectly, downstream of these UPR bZIP-TFs.

## Discussion
ER stress resolution is critical for the survival of the cell and the growth of the entire organism. Owing mostly to the fact that functional studies of the UPR have been mainly conducted in unicellular organisms and cultured cells[2,50], the fundamental question of how tissue growth is controlled by the UPR is still largely unanswered. In this work, we undertook a multi-omics approach based on combining RNA-seq, ChIP-seq, and cistrome analyses into functional gene network analyses in Arabidopsis. Our approach has allowed dissecting the gene regulatory networks operated by bZIP28 and bZIP60 and identified their downstream genes that are functionally involved in organ growth during ER stress recovery, paving the ground to UPR studies also in other multicellular organisms.

Earlier studies indicated that individually bZIP28 and bZIP60 are functionally redundant in gene expression under adaptive UPR[28] and in survival to chronic ER stress[26,34]. However, a quantitative analysis on a handful of UPR genes in adaptive UPR suggested the existence of an overlapping transcriptional regulation by bZIP28 and bZIP60, and a different degree of dependence of gene expression on either UPR bZIP-TF[30]. The same study also indicated that in conditions of ER stress recovery the *bzip28-2* and *bzip60-2* loss-of-function mutants had reduced growth restoration of the primary root compared to Col-0, implying that these UPR bZIP-TFs control the expression of overlapping and unique genes involved in stress recovery and associated organ growth restoration. Our work extends these observations by providing experimental evidence for partially overlapping gene regulatory activities of bZIP28 and bZIP60 on a genome-wide scale. In addition, we established that bZIP28 and bZIP60 temporally regulate gene-regulatory modules for distinct biological processes in the progress of ER stress resolution. Such a temporal gene regulation is likely achieved by the combinatorial effects of shared and unique regulatory activities of bZIP28 and bZIP60 with other TFs or transcriptional regulators. This hypothesis is supported by multiple lines of evidence. First, non-canonical UPR CREs were significantly enriched in the promoters of DDEGs (Fig. 2d), indicating that most likely bZIP28 and bZIP60 directly regulate other TFs that regulate genes in the pink module. Indeed, 16 genes among 354 pink module genes encode TFs, suggesting potential transcriptional cascade effects. Another possibility is

that bZIP28 and bZIP60 generate an environment or conditions in which the expression of these genes is activated (e.g., disruption of metabolic signaling). Second, Our characterization of the UPR-specific binding peaks of bZIP28 and bZIP60 showed an exclusive enrichment of *de novo* motifs statistically similar with DNA-binding motifs of TFs involved in other stress pathways (e.g., MYB96: abscisic acid signaling[51], WRKY75: phosphate deprivation and senescence[52–54], PHL4: phosphate starvation[55]). It is intriguing that not only the genes encoding MYB96 and WRKY75, which were predicted to be coregulators of bZIP28 (Supplementary Fig. 4g, h), were differentially regulated during ER stress, their expression changes were even significantly stronger in *bzip60-2* (Supplementary Fig. 4i). Moreover, the UPR-specific binding peaks of both bZIP28 and bZIP60 had a strong enrichment of ERSE-I, for the binding of both UPR bZIP-TFs, as well as an identical temporal enrichment pattern of bZIP28- and bZIP60-binding signals (Supplementary Fig. 4c–f). We also found distinctive GO terms enriched for the exclusive binding targets of the two UPR bZIP-TFs (i.e., biotic stress for bZIP28 and abiotic stress for bZIP60) (Supplementary Fig. 4j). Therefore, our findings suggest that bZIP28 and bZIP60 interface with gene regulation in the UPR and other stress pathways potentially via coregulation with other TFs. We speculate that this may facilitate resource allocations from other pathways to respond to ER stress and restore growth in specific growth conditions. Activation of the plant UPR in response to biotic and biotic stress has been verified[56–59], but the underlying signaling pathways are largely unknown. The identification of TFs operating in other stress pathways in recovery from ER stress presented in this work provides the opportunity to define the intersection of ER stress with other forms of stress at a mechanistic level.

Although it is known that bZIP28 and to a minor extent bZIP60[30] control necessary pathways for primary root growth, prior to our work, their effectors were not known. Because growth restoration impinges on coordinating a large number of processes, including cell division and elongation, tissue formation, and differentiation[60], we expected that such effectors would be TFs, protein transporters, cell wall modifying enzymes, and many other proteins involved in growth. To rationalize which of the effectors could have a functional role in organ growth in ER stress resolution, we resolved in generating gene regulatory networks. Our network analyses revealed that the temporally regulated combinatorial effects of overlapping and distinctive roles between the two UPR pathways are based on the broad array of coexpression modules specialized for distinct biological processes. The analyses predicted that one of such modules, the pink module, is populated by genes involved in root organogenesis and that COW1 and YIP1a function as two regulatory-hub genes. We functionally validated this prediction by demonstrating that the loss of COW1 and Yip1a partially restore the reduced root growth phenotype of the loss-of-function mutations of *bZIP28* and *bZIP60* in ER stress recovery. Furthermore, the loss of COW1 in the *bzip28-2 bzip60-1* background unambiguously reverted the lethal phenotype of this UPR bZIP-TF double null mutant. While this epistatic relationship underscores that COW1 serves as a life-or-death effector in the UPR, our data do not exclude a possibility that other gene hubs in the organogenesis module may function partially redundantly or independently from the hub genes characterized in this work. Interestingly, null mutants of yeast phosphatidylinositol transfer protein (Sec14p), a COW1 homolog, display a lethal phenotype[61] which is rescued by expression of COW1 lipid-binding domain[62]. Defects in SEC14 proteins also cause human diseases, including retinal degeneration and cancer[63], which are conditions linked to the UPR[64]. In addition to the pink module, other coexpression modules found here are a valuable premise to broaden our view of the impact of the UPR

on other important biological processes. For example, the brown and yellow modules are strongly associated with biotic stress (Fig. 3b), supporting the previous report that the UPR is modulated by key regulators in biotic stress[21,56].

In conclusion, our work addressed the fundamental open question of how growth programs are coordinated during ER stress recovery. Through a multi-omics pipeline that we developed in the model plant Arabidopsis, we identified temporal gene regulatory dynamics of two master transcriptional regulators in ER stress recovery, pinpointed the regulatory-hub genes in a gene regulatory network underlying organ growth, and functionally validated effectors predicted in our network analyses. Our work provides critical insights on the gene regulatory landscape during ER stress recovery with broad applicability across multicellular eukaryotes.

## Methods

**Plant materials**. *A. thaliana* ecotype Columbia-0 (Col-0) was used as the WT control. The following mutants and transgenic lines, which are in Col-0 background except for *bZIP60pro:GFP-bZIP60* (*bzip60-2* background), were used in this study: *bzip28-2* (SALK_132285), *bzip60-2* (SAIL_283_B03), *bzip28-2 bzip60-1* (SALK_132285 SALK_050203), *ire1a ire1b* (WiscDsLox420D09 SAIL_238_F07), *ire1a ire1b bzip60-2*, *35Spro-GFP-bZIP28* and *bZIP60pro:GFP-bZIP60*, *cow1-3* (SALK_002124), *lrx1-4* (SALK_057038), *yip1a-1* (SALK_008449), *yip1a-2* (GK-087C09-012440), and *nac103* (SALK_053122). All T-DNA mutants and high-order mutants used in this study were confirmed to be homozygous before the analyses.

**Plant growth condition**. Surface-sterilized seeds were plated on half-strength Linsmaier Skoog (LS) medium (Caisson Labs, Ontario, Canada) supplemented with 1% sucrose (Sigma-Aldrich, St. Louis, MO, USA), and 1.2% Agar (Acumedia, Lansing, MI, USA). Appropriate antibiotics was also added for screening transgenic lines. After stratification in the dark at 4 °C for 2 days, seeds were transferred to a controlled growth chamber with 80 µmol m$^{-2}$ s$^{-1}$ under 16 h light:8 h dark with 22 °C.

**ER stress assays**. For ER stress recovery assay, 5-day-old seedlings were transferred to half-strength LS liquid buffer containing either 0.5 µg/mL Tm (Sigma-Aldrich, St. Louis, MO, USA) or DMSO-only as mock, and treated for 6 h. After the drug treatment, seedlings were transferred to 100 × 100 square Petri dishes with grid (Fisher Scientific, Hampton, NH, USA) containing half-strength LS medium (Caisson Labs, Ontario, Canada) supplemented with 1% sucrose (Sigma-Aldrich, St. Louis, MO, USA), 1.2% Agar (Acumedia, Lansing, MI, USA). Each square plate was split into two portions, of which each side accommodated either Tm-treated seedlings or DMSO-treated seedlings, which served as a biological replicate (*n* = 6 per replicate). Then, the seedlings grew vertically under the normal growth condition. For adaptive ER stress assay, 5-day-old seedlings were transferred to half-strength LS liquid buffer containing either 0.5 µg/mL Tm (Sigma-Aldrich, St. Louis, MO, USA) or DMSO as mock, and treated for 12 and 24 h. For chronic ER stress assay, seeds were plated on half-strength LS medium (Caisson Labs, Ontario, Canada) supplemented with 1% sucrose (Sigma-Aldrich, St. Louis, MO, USA), 1.2% Agar (Acumedia, Lansing, MI, USA), and 15, 25, 50 ng/mL Tm (Sigma-Aldrich, St. Louis, MO, USA) or DMSO only as mock.

**Phenotyping analysis**. For measuring the primary root length of seedlings, plates were photo-scanned at the same time of the day. The length of primary roots was analyzed with ImageJ software (https://imagej.nih.gov). For measuring dry weight, six whole seedlings of each genotype for each treatment were collected 8 days after stress (DAS) and pooled as a biological replicate (*n* = 5). The dry weight was measured after desiccation for 24 h at 80 °C. All experiments were independently replicated with consistent results.

**LC/MS/MS for Tm measurements**. Following the 6 h of initial treatment with Tm or DMSO, 5-day-old seedlings were subjected to either ER stress recovery (12 and 24 h) or adaptive ER stress assay (12 and 24 h), as detailed above. Tm is a mixture of four homologs A, B, C, and D (Sigma-Aldrich, St. Louis, MO, USA). The Tm 079M4004V batch (cat# T7765) used in this experiment had 8.57% Tm A, 29.91% Tm B, 47.91% Tm C, and 12.18% Tm D. Approximately 30 seedlings were collected for each treatment at each time-point as one of three biological replicates, weighed and then ground in liquid nitrogen. Tm is readily soluble in 80% methanol (https://www.sigmaaldrich.com), which efficiently penetrates plant cell walls/membranes to extract many metabolites[65–67]. Therefore, each ground sample was incubated overnight in 1 mL methanol-based Extraction Buffer (0.1% formic acid, 80% methanol, and 100 nM telmisartan, as an internal standard across samples). Samples were shortly vortexed and centrifuged at 12,000 × *g* at 4 °C for 10 min. The supernatant of each sample was transferred to a new microcentrifuge tube for the

measurement. A standard curve of Tm was prepared by serial dilution in Extraction Buffer to give total Tm concentrations of 0.00122, 0.00488, 0.01953, 0.07813, 0.3125, 1.25, and 5 μM. The endogenous level of Tm contents (A, B, C, and D) in each sample was quantified by LC/MS/MS using a Waters Quattro Premier XE UPLC/MS/MS at the Research Technology Support Facility Mass Spectrometry and Metabolomics Core at Michigan State University. A 10 μL volume of extract was injected onto a Supelco Ascentis Express C18 HPLC column (2.1 × 50 mm, 2.7 μm particle size) maintained at 50 °C. Compounds were separated according to the following gradient run at 0.4 mL/min: initial conditions were 90% mobile phase A (water + 0.1% formic acid) and 10% mobile phase B (methanol) and held for 1 min, ramp to 50% A/50% B at 2 min, then ramp to 100% B at 3.5 min, followed by a ramp back to 90% A/10% B at 4 min and hold at the initial conditions until 5 min. Compounds were ionized by electrospray ionization operating in negative ion mode. The capillary voltage was set to 1.0 kV, the source temperature was 120 °C, the desolvation temperature was 350 °C, the cone gas low was 50 L/h, and the desolvation gas flow was 800 L/h. MS/MS data were acquired using a multiple-reaction-monitoring (MRM) method using the following parameters: Tm A (815.2 > 294.1), Tm B (829.2 > 308.1), Tm C (843.36 > 322.1), and Tm D (857.2 > 336.1), telmisartan (513.2 > 287). All compounds used a cone voltage of 38 V and collision energy of 38 V and dwell time of 0.1 s. Peak areas were integrated using the Quanlynx tool in Masslynx, and concentrations were calculated using the calibration curve with telmisartan used as the internal standard then normalized to the fresh weight of the extracted plant material.

**RNA extraction and qRT-PCR analysis**. Whole seedlings were harvested at 0, 12, and 24 h of ER stress recovery in three biological replicates ($n = 12$ per each replicate), and immediately frozen in liquid nitrogen. The frozen samples were ground to a fine powder in liquid nitrogen using a Retch MM400 Mixer Mill with zirconium oxide balls. Total RNA was extracted using the NucleoSpin RNA Plant kit (MACHEREY-NAGEL, Düren, Germany) according to the manufacturer's instructions. cDNA was synthesized from 1 μg of DNaseI-treated total RNA using iScript cDNA Synthesis Kit (BIO-RAD, Hercules, CA, USA) according to the manufacturer's instruction. For qRT-PCR, Fast SYBR Green Master Mix (Applied Biosystems, Foster City, CA, USA) was used in the presence of gene-specific primers and template cDNAs in an ABI7500 (Applied Biosystems, Foster City, CA, USA).

**RNA-seq analysis**. RNA-seq libraries were constructed using the Illumina TruSeq Stranded mRNA Library (Illumina, San Diego, CA, USA) and sequenced in single-end mode on the Illumina HiSeq4000 platform (50-nt) at the Research Technology Support Facility Genomics Core at Michigan State University. The quality of raw reads was evaluated using FastQC (version 0.11.5). Reads were cleaned for quality and adapters with Cutadapt (version 1.8.1)[68] using a minimum base quality of 20 retaining reads with a minimum length of 30 nucleotides after trimming. Quality-filtered reads were aligned to the Col-0 reference genome (TAIR10) using Bowtie (version 2.2.4)[69] and TopHat (version 2.0.14)[70] with a 10-bp minimum intron length and 15,000-bp maximum intron length. Fragments per kilobase exon model per million mapped reads (FPKM) were calculated using TAIR10 gene model annotation with Cufflinks (version 1.3.0)[71]. Per-gene read counts were measured using HTSeq (version 0.6.1p1)[72] in the union mode with a minimum mapping quality of 20 with stranded = reverse counting. Differential gene expression analysis was performed in each sample relative to the mock control using DESeq2 (version 1.16.1)[73] within R (version 3.4.0). Genes of which the total count is <100 were not included in the analysis. DEGs were obtained based on adjusted $P$ value <0.01 and absolute $Log_2FC > 1$. Among DEGs identified in one or more genotypes at one or more time-points ($n = 6670$), DDEGs were obtained based the pairwise comparison of each DEG between Col-0 and *bzip28-2*, Col-0 and *bzip60-2*, or *bzip28-2* and *bzip60-2* with two-tailed Student *t*-test $P < 0.01$. For visualization purposes, tdf files of each replicate file were generated using igv tools (version 2.3.26) with the command "count" and loaded to Integrative Genome Browser (version 2.5.0)[74]. GO enrichment analysis was performed using agriGO (version 2.0) (http://systemsbiology.cau.edu.cn/agriGOv2/)[75] with a false-discovery rate adjusted $P < 0.05$ (hypergeometric test) as a cutoff. Biological process GO categories were analyzed and the heatmap of GO analysis was produced using R package ggplot2. To validate the RNA-seq profiling, we performed quantitative RT-PCR (qRT-PCR) for 11 genes including seven known UPR biomarker genes and four downstream genes identified or will be identified below. We generated a total of 93 data points ($Log_2[Tm/DMSO]$) across the time-points and genotypes (Supplementary Fig. 2b), which significantly correlated with those measured by the RNA-seq analysis (PCC = 0.947, $P < 2.2 \times 10^{-16}$).

**Coexpression analysis**. Coexpression network was built using the WGCNA R package[36]. All DEGs obtained from the DESeq2 analysis with the strict criteria (total read count across six samples [three for Tm-treated and three for DMSO-treated] >100, adjusted $P < 0.01$ and absolute $Log_2$-transformed fold change >1) were considered. $Log_2$-transformed DESeq2-normalized fold-changes (Tm/DMSO) served as input. Through empirical optimizations, a soft-power threshold of 18 was selected to create a signed network of a Spearman correlated matrix. Topographical overlap matrices (TOMs) were constructed using TOMsimilarityFromExpr() function with default parameter. The TOM scores were used as edge weights in the analysis. Coexpression modules were constructed through hierarchical clustering the TOM distance using flashClust() function with (method = "average"). Modules were derived using the cutreeDynamic() function with 25 of minimum module size, and similar modules were merged into a single module using mergeCloseModules() function with (cutHeight = 0.05). Network visualization was achieved using the Cytoscape software (version 3.6.1) with a cutoff of the weight parameter obtained from the WGCNA, set at 0.2.

**Chromatin immunoprecipitation (ChIP) assays, ChIP-qPCR, ChIP-seq library preparation**. ChIP was performed using a modified protocol based on general methods previously described[76]. Whole seedlings harvested at 0, 12, and 24 h of ER stress recovery in three biological replicates were completely submerged in freshly prepared pre-chilled crosslinking buffer (0.4 M Sucrose, 10 mM Tris-HCl pH 8.0, 1 mM EDTA, 1 mM PMSF, and 1% Formaldehyde) in 50 mL conical tubes. The samples were subjected to a vacuum infiltration for 15 min and another for 5 min with glycine (125 mM final concentration). After the crosslinking buffer was removed, the crosslinked tissues were briefly rinsed with prechilled sterilized water, dried by gently blotting between paper towels, immediately frozen in the liquid nitrogen, and stored at −80 °C for the next steps. The frozen tissues were ground to a fine powder in liquid nitrogen using a prechilled motor and pestle. The tissue powder was homogenized in 30 mL of pre-chilled nuclei isolation buffer A (0.4 M Sucrose, 10 mM Tris-Cl pH 8.0, 1 mM PMSF, 5 mM β-Mercaptoethanol and 1× Protease inhibitor cocktail [Sigma-Aldrich, St. Louis, MO, USA; cat# P2714]) at 4 °C for 30 min. Homogenized tissue powder was filtered through Miracloth and then centrifuged at 4 °C with 2880 g for 20 min. After centrifugation, the supernatant was discarded and pellets were resuspended in 1 mL of pre-chilled nuclei isolation buffer B (0.25 M Sucrose, 10 mM MgCl₂, 1% Triton X-100, 10 mM Tris-Cl pH 8.0, 1 mM PMSF, 5 mM β-Mercaptoethanol and 1× Protease inhibitor cocktail [Sigma-Aldrich, St. Louis, MO, USA; cat# P2714]). The suspension was centrifuged at 4 °C with 12,000 g for 10 min. After centrifugation, the supernatant was discarded, and pellets were resuspended in 300 μL of pre-chilled nuclei isolation buffer C (1.7 M Sucrose, 2 mM MgCl₂, 0.15% Triton X-100, 10 mM Tris-Cl pH 8.0, 1 mM PMSF, 5 mM β-Mercaptoethanol and 1× Protease inhibitor cocktail [Sigma-Aldrich, St. Louis, MO, USA; cat# P2714]). The suspension was added to the top of 1.5 mL cold nuclei isolation buffer C and centrifuged at 4 °C with 16,000 g for 1 h. After centrifugation, the pellet was resuspended in 310 μL of pre-chilled nuclei lysis buffer (50 mM HEPES, pH 7.5, 150 mM NaCl, 1 mM EDTA, 1 mM PMSF, 1% SDS, 0.1% Na deoxycholate, 1% Triton X-100, 1 μg/mL Pepstatin A [Sigma-Aldrich, St. Louis, MO, USA; cat# P5318] and 1× Protease inhibitor cocktail [Sigma-Aldrich, St. Louis, MO, USA; cat# P2714]). After isolation, chromatin was fragmented using Covaris M220 sonicator (Covaris, Woburn, MA, USA) with settings of three cycles of PIP-50, duty factor 20% time-70s at 4 °C. Immunoprecipitation (IP) was performed with a polyclonal anti-GFP antibody (Abcam, Cambridge, UK; cat# ab290; 1/200 dilution). For each ChIP sample, a mock (no antibody) and input (no IP) were included for control experiments. About 2 μL purified DNA of ChIP, mock and input samples, which was diluted by fourfold, was used for qPCR analysis in an ABI7500 machine (Applied Biosystems, Foster City, CA, USA) using Fast SYBR Green Master Mix (Life Technologies, Carlsbad, CA, USA). The enrichment of amplicons from ChIP DNA samples was normalized relative to one of the corresponding input DNA samples. Three technical replicates were assayed for each of the three biological replicates. The list of primers used in ChIP-qPCR is provided in Supplementary Table 1. The final purified ChIP and input DNAs were quantified using the Qubit fluorometer (Thermo Fisher Scientific, Carlsbad, CA, USA), and subjected to ChIP-seq libraries construction in two biological replicates using the NEBNext Ultra II DNA Library Prep Kit (New England BioLabs, Beverly, MA, USA) according to manufacturer's protocol. The suitable size distribution of libraries was confirmed using the 2100 Bioanalyzer (Agilent, Santa Clara, CA, USA). Multiplexed libraries were sequenced in single-end mode on the Illumina HiSeq4000 platform (50-nt) at the Research Technology Support Facility Genomics Core at Michigan State University.

**ChIP-seq data analysis**. The guideline and analysis pipeline provided by the ENCODE project[42] was adapted to our ChIP-seq analysis with several modifications optimized for the dataset. The quality of raw ChIP-seq reads was evaluated using FastQC (version 0.11.5). Reads were cleaned for quality and adapters with Cutadapt (version 1.8.1)[68] using a minimum base quality of 20 retaining reads with a minimum length of 30 nucleotides after trimming. Quality-filtered reads were aligned to the Col-0 reference genome (TAIR10) using Bowtie (version 1.1.2)[77] with parameters "-n 2 -m 3 -k 1 –threads 7 –best –chunkmbs 256 -q". Duplicated reads were removed using Samtools (version 1.8)[78]. Peak calling was performed using MACS2 (version 2.1.2)[79] in individual samples with input samples pooled for each of DMSO- and Tm-treated samples and with a relaxed threshold of $P$ value (−p = 1e-2), as recommended by the IDR pipeline (https://sites.google.com/site/anshulkundaje/projects/idr). Peaks across replicates with an IDR <0.05 were retained for further analysis. To obtain binding peaks with high-confidence, we applied two parameters to IDR-filtered peaks; (1) a peak called in Tm samples that was overlapped with a peak in the corresponding DMSO-treated samples by >30% was eliminated, (2) Among the peaks that were overlapped with peaks in DMSO-treated sample, if its $P$ float (eighth column in the IDR output file) in the

Tm-treated sample was higher than the corresponding peak in the DMSO-treated sample by greater than threefold, the peak was retained and named as a UPR-specific binding peak. UPR-specific binding peaks obtained at each time-point were merged into a single list for further analysis and were annotated using the ChIP-seeker and the GenomicFeatures R Package (version 3.4.0)[80]. For visualization purposes, bigwig files (using pooled data across biological replicates) were generated by deepTools suite (https://deeptools.readthedocs.io/en/develop/)[81] with the command "bamCoverage"; read coverage was normalized as RPKM (Reads Per Kilobase per Million reads). ChIP-seq tracks were visualized in Integrative Genome Browser (version 2.5.0)[74]. ChIP-seq metaplot was generated from the merged single file by deepTools suite with the commands "bamCompare" and "plotProfile". UPR-specific binding peaks were mapped to the vicinity of a coding sequence (<1-kb), generating a total of 440 bZIP28- and 356 bZIP60-bound genes, respectively. GO enrichment analysis was performed as described in RNA-seq analysis.

**Motif discovery and enrichment analysis**. All motif analyses presented in this study were performed using various tools of the MEME suite (version 5.0.5) (http://meme-suite.org) with default parameters and modifications indicated elsewhere. Random model letter frequencies were calculated based on GC content in the Arabidopsis genome (https://genomevolution.org/wiki/index.php/Plant_genome_GC_content) and used for the following analyses. 1-kb upstream sequence of the transcription start sites (TSS), hereafter called 1-kb promoter, in the TAIR10 Arabidopsis genome annotation was used. Enrichment of canonical CREs in response to ER stress on the promoters of DDEGs was investigated using AME[82]. For this analysis, one mismatch was allowed for each CRE except for UPRE-I. De novo motif discovery in 1-kb promoter of DDEGs was performed using MEME[83]. To eliminate any de novo motifs not specific to DDEGs, 201, which is the median number of DDEGs across all pairwise comparisons, DEGs were randomly selected and served as a control for the analyses. The similarity of enriched motifs with the Plant Cistrome Database[84] was assessed using TOMTOM[85]. De novo motif discovery in UPR-specific binding peaks of bZIP28 and bZIP60 from the ChIP-seq analyses was carried out using MEME-ChIP[86]. For this analysis, UPR-specific binding peaks obtained at each time-point were combined, creating union sets of peaks for bZIP28 and bZIP60. Then, 250-bp sequence upstream and downstream of the summit in each of the combined peaks was obtained from the TAIR10 Arabidopsis genome annotation; for peaks that had multiple sub-peaks, one that had the highest of -Log$_{10}$P was selected. The enrichment distribution of ERSE-I in UPR-specific binding peaks was analyzed using CentriMo[87].

**Statistics and reproducibility**. For experimental analyses, a two-tailed Student's *t*-test was used to obtain *P* values in this study using the R function and values of $P < 0.05$ were considered statistically significant. The number of samples is indicated in the respective Figure legends: *n* (number of biological replicates) = 3 (~50 seedlings per replicate), Fig. 1b; *n* = 3 (12 seedlings per replicate), Fig. 1c and Supplementary Fig. 3a, d, g; *n* = 5 (6 seedlings per replicate), Fig. 1d and Supplementary Figs. 1b, 3b; *n* = 3–5 (6 seedlings per replicate), Fig. 5a; *n* (number of biological replicates) = 4–5 (3–5 seedlings per replicate), Fig. 5b; *n* = 10 (2 seedlings per replicate), Supplementary Fig. 1c; *n* = 3 (~100 seedlings per replicate), Supplementary Fig. 2f; *n* = 6–8, Supplementary Fig. 6. No data points were removed from the statistical analyses as outliers. Results were confirmed in at least two independent experiments.

**Reporting summary**. Further information on research design is available in the Nature Research Reporting Summary linked to this article.

## Data availability

Data supporting the findings of this study are available within this paper and its Supplementary Information files. RNA-seq and ChIP-seq data supporting the finding of this study have been deposited in the NCBI Gene Expression Omnibus and are accessible through the GEO series accession code GSE146723. The source data, which consists of an excel spreadsheet, are provided in Supplementary Data 6.

## Code availability

The bioinformatic scripts used in this study are available on GitHub (https://github.com/DaeKwan-Ko/UPR-GRN.git).

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

## Acknowledgements
This study was supported primarily by the National Institutes of Health (R35-GM136637) with contributing support from by the Great Lakes Bioenergy Research Center, US Department of Energy, Office of Science, Office of Biological and Environmental Research (DE-SC0018409), Chemical Sciences, Geoscience, and Biosciences Division, Office of Basic Energy Sciences, Office of Science, US Department of Energy (DE-FG02-91ER20021), and MSU AgBioResearch (MICL02598). We thank the Research Technology Support Facility Genomics Core and Mass Spectrometry and Metabolomics Core facilities at Michigan State University for the next-generation sequencing and quantitative LC/MS/MS, respectively. We also thank Cristina Ruberti for preparing samples for RNA-seq analysis.

## Author contributions
D.K.K. and F.B. conceived the original experiment design and research plan; F.B. supervised the project and experiments; D.K.K. performed experiments and data analyses; D.K.K. and F.B. wrote the article.

## Competing interests
The authors declare no competing interests.
