## [Peer Review File · Communications Biology]

Reviewers' comments:

Reviewer #1 (Remarks to the Author):

The manuscript "Genomics-enabled identification of growth effectors for proteotoxic ER stress recovery" by Ko and Brandizzi focuses on the downstream regulators of two major ER stress-related transcription factors, bZIP60 and bZIP28. Specifically, the authors used the genetics approach, performed a Tm-based recovery assay, and found that functional loss of bZIP60 and bZIP18 did not result in growth recovery defects from ER stress and these mutants displayed phenotypes similar to that of the wild-type plants. Besides additional recovery assays, the authors performed RNA-seq in WT and mutants corresponding to bZIP60 and bZIP18 samples collected at 0 h, 12 h, and 24 h after ER stress application during the stress recovery phase. This led to the discovery of a set of DEGs in a bZIP60- and bZIP18-specific manner. Following that the authors performed promoter surveys of the DEGs and identified potential direct gene regulation by these two master TFs. Next, the authors performed WGCNA to identify co-expressed modules and identify hubs. Subsequently, the authors performed ChIP-Seq on a time course equivalent to the one described above for RNA-seq. By integrating the ChIP-seq and transcriptomic analyses, the authors inferred bZIP28- or bZIP60-dependent gene regulation. The authors further addressed whether other TFs in conjunction with these two major UPR players participate in the regulation of ER stress genes. This led to the identification of additional TFs that might play a role in the regulation of ER stress genes. Overall, this analysis identifies a total of 440 bZIP28- and 356 bZIP60-bound genes. This also led to the discovery of Can of Worms1 (COW1), which encodes a phosphatidylinositol transfer protein essential for root hair tip growth and other developmental processes. Additional hubs are Leucine-rich repeat/Extensin1 (LRX1, ANAC103, and Yip1 family protein 1a (YIP1a). The authors employed reverse genetics and performed recovery assays on the mutants corresponding to these hubs. While *irx1-4* showed no growth alteration, *cow1-3*, *yip1-1*, and *yip1a-2* exhibited a significant reduction in the relative growth rate of the primary root compared to Col-0. The authors tested epistatic relationships between these hub genes and two UPR TFs by generating higher-order mutants. The authors discovered that that *cow1-3 bzip28-2* and *yip1a-1 bzip28-2* displayed a significantly higher relative root growth recovery compared to *bzip28-2*.

Overall, this study is well-thought-out and will be a step forward in solving the puzzle of UPR in plants. However, I have a few points that need clarity.

1- The authors reasoned why they selected the pink module based on a unique expression profile of the eigengene in each genotype and their enrichments in root growth GO terms. Integration of ChIP-seq data with pink modules provided the direct or indirect regulation status of the genes by these two TFs. However, the choice of the genes for genetics analyses and downstream characterization were based on network centrality analysis, high degree (hubness) to be more specific. So, it's not clear what information ChIP-seq provided in this context besides that ANAC103 was bound directly by bZIP60. So, how the LRX1 and YIP1a are regulated? Are they co-expressed with bZIP60 and/or bZIP28? Are these four genes differentially regulated in the mutants corresponding to bZIP28 and bZIP60? Are bZIP28 and bZIP60 part of the pink module? Did the author find bZIP28 and bZIP60 as hubs? Also, what are the reasons behind selecting these four hubs among other hubs? Would other hubs being part of the root development module also display root phenotypes? Could the author comment on any known hub or non-hubs genes that have previously been shown to be involved in root morphology? It is suggested that the authors also display the network according to the degree of nodes (size of the nodes should depict the degree). Alternatively, they should display hubs in a different color.

2- In figure 2, the authors predicted additional TFs, that may participate in cellular stress. It's not clear whether these predicted TFs that may work in concert with bZIP60 and bZIP28 and how they contribute to TM-induced cellular stress.

Reviewer #2 (Remarks to the Author):

Although the unfolded protein response (UPR), a conserved cytoprotective signaling pathway, plays a pivotal role in restoring growth during endoplasmic reticulum (ER) stress, the mechanism by which the UPR modulators bZIP28 and bZIP60 contribute to the restoration has yet to be determined. The submitted manuscript set up an ER stress recovery assay using bzip mutants, carried out RNA-seq and ChIP-seq analyses, and identified differentially expressed genes (DEGs) among genotypes. The authors further demonstrated the gene regulatory network in which bZIP28 and bZIP60 are involved in partially-overlapping but distinct stress response pathways during ER stress recovery. They also identified novel genes that are important for growth recovery from ER stress by genetic analysis. This work adds interesting information on how two UPR modulators contribute to growth recovery from ER stress and will be of interest to not only plant researchers but also researchers working on animal systems. I have the following points to be addressed before publication.

ER stress recovery data shown in Fig.5 would need more discussion. It seems to be that the authors do not give sufficient explanation about the different effects of cow1 and yip1a mutation on root growth during ER stress recovery among wild-type and mutant plants (i.e. cow1 and yip1a single mutants are more sensitive to Tm than Col-0 but their effects are reversed in both bzip28 and bzip60 background). For example, Fig.4 shows that COW1 expression is clearly altered in bzip28 but unaffected in bzip60, so if cow1 is more sensitive to Tm than Col-0 and cow1 bzip28 is less sensitive than bzip28, then shouldn't cow1 bzip60 be more sensitive to Tm than bzip60? The same hold true for YIP1a. If I didn't understand the authors' interpretation, I apologize and will be please to read the authors' reply.

Other points:

Fig. 5B: If relative root growth in y-axis is calculated by dividing a primary root length of Tm-treated seedlings by that of DMSO-treated seedlings, no data point should be below zero. Also, no data point should go over 1 unless Tm stimulates root growth.

Fig. 1E: The red bar for Tm-treated Col-0 should be on the bottom of the primary roots.

We appreciate the constructive comments of the reviewers. As detailed in our response below, we have addressed all of the reviewers' concerns. We are grateful for the comments expressed by the reviewers because they helped us improve our work.

Reviewer #1 (Remarks to the Author):

The manuscript "Genomics-enabled identification of growth effectors for proteotoxic ER stress recovery" by Ko and Brandizzi focuses on the downstream regulators of two major ER stress-related transcription factors, bZIP60 and bZIP28. Specifically, the authors used the genetics approach, performed a Tm-based recovery assay, and found that functional loss of bZIP60 and bZIP18 did not result in growth recovery defects from ER stress and these mutants displayed phenotypes similar to that of the wild-type plants. Besides additional recovery assays, the authors performed RNA-seq in WT and mutants corresponding to bZIP60 and bZIP18 samples collected at 0 h, 12 h, and 24 h after ER stress application during the stress recovery phase. This led to the discovery of a set of DEGs in a bZIP60- and bZIP18-specific manner. Following that the authors performed promoter surveys of the DEGs and identified potential direct gene regulation by these two master TFs. Next, the authors performed WGCNA to identify co-expressed modules and identify hubs. Subsequently, the authors performed ChIP-Seq on a time course equivalent to the one described above for RNA-seq. By integrating the ChIP-seq and transcriptomic analyses, the authors inferred bZIP28- or bZIP60-dependent gene regulation. The authors further addressed whether other TFs in conjunction with these two major UPR players participate in the regulation of ER stress genes. This led to the identification of additional TFs that might play a role in the regulation of ER stress genes. Overall, this analysis identifies a total of 440 bZIP28- and 356 bZIP60-bound genes. This also led to the discovery of Can of Worms1 (COW1), which encodes a phosphatidylinositol transfer protein essential for root hair tip growth and other developmental processes. Additional hubs are Leucine-rich repeat/Extensin1 (LRX1, ANAC103, and Yip1 family protein 1a (YIP1a). The authors employed reverse genetics and performed recovery assays on the mutants corresponding to these hubs. While *irx1-4* showed no growth alteration, *cow1-3*, *yip1-1*, and *yip1a-2* exhibited a significant reduction in the relative growth rate of the primary root compared to Col-0. The authors tested epistatic relationships between these hub genes and two UPR TFs by generating higher-order mutants. The authors discovered that that *cow1-3 bzip28-2* and *yip1a-1 bzip28-2* displayed a significantly higher relative root growth recovery compared to *bzip28-2*.

Overall, this study is well-thought-out and will be a step forward in solving the puzzle of UPR in plants. However, I have a few points that need clarity.

1- The authors reasoned why they selected the pink module based on a unique expression profile of the eigengene in each genotype and their enrichments in root growth GO terms. Integration of ChIP-seq data with pink modules provided the direct or indirect regulation status of the genes by these two TFs. However, the choice of the genes for genetics analyses and downstream characterization were based on network centrality analysis, high degree (hubness) to be more specific. So, it's not clear what information ChIP-seq provided in this context besides that ANAC103 was bound directly by bZIP60.

This is a good point. Through the ChIP-seq analysis, we found that the majority of genes in the pink module are not direct targets of bZIP28 and/or bZIP60. This is partially reflected by the presence of multiple CREs enriched in the promoters of DDEGs shown in Fig. 2D. Most likely, bZIP28 and bZIP60 directly regulate other TFs that regulate genes in the pink module. Indeed, 16 genes among 354 pink module genes encode TFs, suggesting potential transcriptional cascade effects. Another possibility is that bZIP28 and bZIP60 generate an environment or conditions in which the expression of these genes is activated (e.g., disruption of metabolic signaling). We have illustrated these possibilities in the revised manuscript.

So, how the LRX1 and YIP1a are regulated? Are they co-expressed with bZIP60 and/or bZIP28?

As also acknowledged by the reviewer, our coexpression network analysis established that the expression of pink module genes is uniquely affected by the absence of either bZIP28 or bZIP60 (Fig. 4A). Thus, those genes are under the control of bZIP28 and/or bZIP60, but not necessarily coexpression

partners. In the revised manuscript, the expression pattern of the selected hub genes is visualized in a separate heatmap in Fig 5B.

Are these four genes differentially regulated in the mutants corresponding to bZIP28 and bZIP60?

Yes, as described above, the expression of the selected hub genes is affected by the absence of bZIP28 and bZIP60 in a temporal manner (Fig. 5B).

Are bZIP28 and bZIP60 part of the pink module? Did the author find bZIP28 and bZIP60 as hubs?

No, they are not because they are the ones that regulate the pink module.

Also, what are the reasons behind selecting these four hubs among other hubs?

The hub genes were selected based on the criteria described in the manuscript (Line 237-239): “We considered two main criteria: ranking in the top $\geq 10\%$ of network degree (i.e., the number of connections) and having a known phenotype of loss-of-function mutations or expected role in roots, cell development and/or ER stress processes”.

Would other hubs being part of the root development module also display root phenotypes? Could the author comment on any known hub or non-hubs genes that have previously been shown to be involved in root morphology?

According to the GO term analysis, root development associated pathways were strongly (the lowest P-value = $4.60E-33$) enriched exclusively among the pink module genes (Fig. 3B). In line with that, many genes including the hubs in the pink module have been experimentally shown to associate with root development and morphology. For example, the hub gene, *COW1*, and non-hub gene, *LRX1*, are known to be associated with root hair morphogenesis (DOI: 10.1371/journal.pgen.1002446). Overexpressing *NAC103* was shown to cause root growth defects (DOI: 10.1111/tbj.12287). Another pink module gene, *PROLINE-RICH PROTEIN-LIKE 1 (AT5G05500)*, which was not characterized in this study, is also known to be associated with root development (DOI: 10.1093/jxb/eru308). We do not exclude the possibility that other potential hub genes play a key role in the pink module and root growth during ER stress recovery. This has been indicated in the revised manuscript.

It is suggested that the authors also display the network according to the degree of nodes (size of the nodes should depict the degree). Alternatively, they should display hubs in a different color.

Thank you for your suggestion. The degree of nodes is now indicated as the size of nodes in the network map (Fig. 4A).

2- In figure 2, the authors predicted additional TFs, that may participate in cellular stress. It's not clear whether these predicted TFs that may work in concert with bZIP60 and bZIP28 and how they contribute to TM-induced cellular stress.

In Fig. 2d, through the *de novo* motif analysis, we found potential cis-regulatory elements (CREs) significantly enriched in the promoters of genes regulated in a genotype-dependent manner. These CREs were significantly similar with specific TFs motifs, indicating that UPR target genes may be controlled by such TFs or transcriptional cascades triggered by bZIP28 and bZIP60. This has been clarified in the revised manuscript.

Reviewer #2 (Remarks to the Author):

Although the unfolded protein response (UPR), a conserved cytoprotective signaling pathway, plays a pivotal role in restoring growth during endoplasmic reticulum (ER) stress, the mechanism by which the UPR modulators bZIP28 and bZIP60 contribute to the restoration has yet to be determined. The submitted manuscript set up an ER stress recovery assay using bzip mutants, carried out RNA-seq and

ChIP-seq analyses, and identified differentially expressed genes (DEGs) among genotypes. The authors further demonstrated the gene regulatory network in which bZIP28 and bZIP60 are involved in partially-overlapping but distinct stress response pathways during ER stress recovery. They also identified novel genes that are important for growth recovery from ER stress by genetic analysis. This work adds interesting information on how two UPR modulators contribute to growth recovery from ER stress and will be of interest to not only plant researchers but also researchers working on animal systems. I have the following points to be addressed before publication.

ER stress recovery data shown in Fig.5 would need more discussion. It seems to be that the authors do not give sufficient explanation about the different effects of *cow1* and *yip1a* mutation on root growth during ER stress recovery among wild-type and mutant plants (i.e., *cow1* and *yip1a* single mutants are more sensitive to Tm than Col-0 but their effects are reversed in both *bzip28* and *bzip60* background). For example, Fig.4 shows that *COW1* expression is clearly altered in *bzip28* but unaffected in *bzip60*, so if *cow1* is more sensitive to Tm than Col-0 and *cow1 bzip28* is less sensitive than *bzip28*, then shouldn't *cow1 bzip60* be more sensitive to Tm than *bzip60*? The same hold true for *YIP1a*. If I didn't understand the authors' interpretation, I apologize and will be please to read the authors' reply.

This is a good point of discussion. Thank you. First of all, we believe that it was difficult to access the expression patterns of the individual hub genes from the heatmaps (Fig. 5b) because the heatmaps display expression patterns of all pink module genes ($n = 354$). This may have generated confusion. To address it, we have provided a heatmap for the selected candidate hub genes (*COW1*, *LRX1*, *YIP1a* and *NAC103*). The revised heatmap it is more clearly possible to see that while there was little change in gene expression across genotypes at 0 h, *COW1* expression was substantially downregulated in both *bzip28-2* and *bzip60-2* (not in Col-0) at 12 h. The downregulation of *COW1* expression was maintained only in *bzip28-2* at 24 h (not in *bzip60-2*). *YIP1a* expression was substantially upregulated in Col-0 and *bzip28-2* (not in *bzip60-2*) at 0 h while the upregulation was maintained in both *bzip28-2* and *bzip60-2* at a different extent at 12 h and 24 h. Because our network analysis is based on "guilt-by-association" (i.e., expression similarity across genotypes and time-points), temporal genetic effects of *bzip28-2* and *bzip60-2* hold this network together in the context of gene expression. This explains the epistatic relationship between *cow1* (or *yip1a*) and *bZIP28/bZIP60*.

Other points:

Fig. 5B: If relative root growth in y-axis is calculated by dividing a primary root length of Tm-treated seedlings by that of DMSO-treated seedlings, no data point should be below zero. Also, no data point should go over 1 unless Tm stimulates root growth.

The typo on the y-axis has been fixed. Thank you for pointing this out.

Fig. 1E: The red bar for Tm-treated Col-0 should be on the bottom of the primary roots.

Revised.

REVIEWERS' COMMENTS:

Reviewer #1 (Remarks to the Author):

The authors addressed all of my comments to my satisfaction. No further changes or revisions are requested.

Reviewer #2 (Remarks to the Author):

The revised manuscript and the authors' response addressed all the concerns raised, and there is no problem for publication.